# 4D In-Situ Microscopy of Aerosol Filtration in a Wall Flow Filter

**DOI:** 10.3390/ma13245676

**Published:** 2020-12-12

**Authors:** Matthew P. Jones, Malte Storm, Andrew P. E. York, Timothy I. Hyde, Gareth D. Hatton, Alex G. Greenaway, Sarah J. Haigh, David S. Eastwood

**Affiliations:** 1Department of Materials, University of Manchester, Manchester M13 9PL, UK; sarah.haigh@manchester.ac.uk; 2University of Manchester at Harwell, Diamond Light Source, Harwell Science & Innovation Campus, Didcot, Oxfordshire OX11 0DE, UK; alex.greenaway@manchester.ac.uk; 3UK Catalysis Hub, Research Complex at Harwell, Rutherford Appleton Laboratory, Harwell Science & Innovation Campus, Didcot, Oxfordshire OX11 0FA, UK; 4Diamond Light Source, Harwell Science & Innovation Campus, Didcot, Oxfordshire OX11 0DE, UK; malte.storm@diamond.ac.uk; 5Johnson Matthey Technology Centre, Blounts Court Road, Sonning Common, Reading RG4 9NH, UK; andrew.york@matthey.com (A.P.E.Y.); timothy.hyde@matthey.com (T.I.H.); gareth.hatton@matthey.com (G.D.H.); 6Department of Mechanical, Aerospace and Civil Engineering, University of Manchester, Manchester M13 9PL, UK

**Keywords:** X-ray micro-CT, synchrotron imaging, novel methods, in-situ, aerosol, filtration, porous media, particulate filter, particulate matter

## Abstract

The transient nature of the internal pore structure of particulate wall flow filters, caused by the continuous deposition of particulate matter, makes studying their flow and filtration characteristics challenging. In this article we present a new methodology and first experimental demonstration of time resolved in-situ synchrotron micro X-ray computed tomography (micro-CT) to study aerosol filtration. We directly imaged in 4D (3D plus time) pore scale deposits of TiO${}_{2}$ nanoparticles (nominal mean primary diameter of 25 nm) with a pixel resolution of 1.6 μm. We obtained 3D tomograms at a rate of ∼1 per minute. The combined spatial and temporal resolution allows us to observe pore blocking and filling phenomena as they occur in the filter’s pore space. We quantified the reduction in filter porosity over time, from an initial porosity of 0.60 to a final porosity of 0.56 after 20 min. Furthermore, the penetration depth of particulate deposits and filtration rate was quantified. This novel image-based method offers valuable and statistically relevant insights into how the pore structure and function evolves during particulate filtration. Our data set will allow validation of simulations of automotive wall flow filters. Evolutions of this experimental design have potential for the study of a wide range of dry aerosol filters and could be directly applied to catalysed automotive wall flow filters.

## 1. Introduction

In 2018, 82% of urban dwellers still breathed air below the standards set by the World Health Organization’s air quality guidelines for particulate matter (PM) [1]. These state that PM levels should not increase above an annual mean of 10 μg/m${}^{3}$ for PM2.5 or 20 μg/m${}^{3}$ for PM10. A key technological factor in arresting PM before it pollutes the urban environment is the installation of particulate filters on the exhaust of road vehicles with combustion engines. Particulate filters are ’wall flow filters’ with channels that run parallel to exhaust gas flow. Channels are blocked alternately at the inlet, then outlet, of neighbouring channels in a ’honeycomb’ monolith. The blocked channel ends force the exhaust gas to flow through the connected pores in the walls of the particulate filter, from an inlet channel to an outlet channel, filtering out particles. It is well reported in literature that during continuous filtration a filter will transition from deep bed filtration (where particles deposit in the interior pores of the filter) to cake filtration (where particles deposit as a layer on the channel surface) [2]. The deposits cause the permeability of the filter to change continuously. An increase in permeability can lead to back-pressure from the exhaust lowering the fuel efficiency of the engine, increasing green house gas emission per mile. Thus, understanding the effects of these pore-scale structural changes, how they choke flow through the filter and how they can be efficiently burnt-off or regenerated is an unmet challenge which requires knowledge of the pore scale 3D distribution of PM deposits over time in particulate filters [3].

Literature studies have applied computational fluid dynamics (CFD) to micro-CT volumes of fresh filters to simulate filtration efficiency [4,5] and other processes [6,7,8,9]. However, pore scale image-based simulations are yet to incorporate the transient nature of the pore structure caused by continuous PM deposition with experimental data. Models have simulated the 3D build-up of deposit [5,10,11,12], however this method for generating the deposit phase is yet to be directly validated by experiments observing the deep bed and pore scale build up of deposit overtime. This is a gap in the literature. In-situ time resolved imaging of surface cake filtration was achieved with optical microscopy by Choi and Lee [13], however optical microscopy can only interrogate surface features, not the deep bed filtration within the filter walls. Scanning Electron Microscopy (SEM) has been widely carried out in order to construct 3D volumes of filters [10,11,12] and fluorescence mapping of deposits in the filter walls has been carried out with SEM energy dispersive X-ray spectroscopy [10,12]. Furthermore, Kamp et al. imaged the structure of the ash layer and PM deposits in pore interiors by employing focused ion beam SEM and atomic force microscopy [14,15]. These destructive techniques make it impossible to investigate the interior of intact operational samples in an in-situ time resolved manner as they require sectioning. A number of studies have used non-destructive micro-CT to examine the internal structure of the filters at a range of different length scales [16,17] and Kamp et al. [9] used high resolution laboratory micro-CT to image the ash PM layer structure and distribution in the internal pores. However, small non-operational samples had to be imaged with long scan times by these laboratory micro-CT techniques.

In-situ, time resolved, and pore scale imaging of aerosol particulate filters would improve understanding of the dynamic changes in the filter’s porous media in-use. Time resolved and pore scale imaging is often applied in geosciences and petrophysics when studying multiphase flow and pore transport phenomena in permeable rock formations [18]. In these geological studies high-speed synchrotron-based X-ray imaging has captured time resolved pore scale events, such as, pore-filling, pore snap-off, coalescence and ganglia dynamics during drainage and imbibition with both steady and unsteady flow conditions [19,20,21]. Whilst 4D laboratory micro-CT experiments are possible [22], sub-minute time resolution and micron scale X-ray imaging requires the high photon flux of synchrotron X-ray computed tomography beamlines, such as the I13-2 imaging branchline at Diamond Light Source, UK [23]. Here, we present our experimental set-up at I13-2 and initial results from our study that imaged the build-up of particulate deposits in the internal pores and on the channel walls of an automotive particulate filter whilst an aerosol was continuously flowed through the sample.

## 2. Materials and Methods

### 2.1. Imaging Method

In order to achieve 4D in-situ imaging of particulate deposition an aerosol flow rig had to be set-up in the Diamond Light Source I13-2 beamline hutch so that aerosol could be flowed through the sample whilst the sample was illuminated with X-rays and rotated for tomography. For fast in-situ imaging, a radiographic projection was taken every 0.18${}^{\circ }$ in an 180${}^{\circ }$ fly scan (1000 projections) with an exposure of 0.01 s. Flat field and dark field images were taken before and after each in-situ run rather than during the scan in order to reduce the time taken per projection. These flats and darks were used in post to correct image aberrations during reconstruction. The imaging was performed using the ‘pink beam’ configuration at I13-2 which utilizes several harmonic peaks from the undulator X-ray source. The ID gap was 5 mm, and a 3.2 mm Al filter was installed before the sample. Thus, the spectrum reaching the sample ranged from around 6 to 35 keV with a peak flux of 25 keV and an integrated photon yield at the sample position approximately 3.72 × 10${}^{12}$ photons/s. A $CdWO_{3}$ scintillator was used with the 1.25× objective, and the 2× and 4× objectives were used with LuAg scintillators to project visible light through the objective to a CMOS camera (pco.edge 5.5) to capture radiographic projections, as in Figure 1 below.

The spectrum was optimized to maintain sufficient X-ray transmission through the sample holder (5 mm thick 3D printed Acrylonitrile Butadiene Styrene) and filter walls while achieving sufficient contrast for the deposits. This set-up required a bespoke 3D printed sample holder, see Figure 1. It was strong enough to prevent deflection during rotation, thin enough to minimize unnecessary beam attenuation and also leak proof to prevent aerosol escape or pressure loss. The filter sample was placed in the sample holder at the thin central section via a door in the side of the holder. The sample holder reduced in width so that flow entered the sample holding region through a small 1 mm × 1 mm inlet. This directed flow through the central channel of the filter sample. Two outlets perpendicular to the flow allowed for a differential pressure measurement across the filter, see P1 and P2 in Figure 1.

### 2.2. Aerosol Flow Apparatus

A Topas SAG 410/U dust generator was used to aerosolize a TiO${}_{2}$ nanopowder (nominal mean primary diameter 25 nm) which was flowed through the sample holder and filter sample by a head pressure provided by an air compressor. This is illustrated below in Figure 2.

The dust generator operated by drizzling powder over a thin rotating dosing ring. A powder ‘scraper’ was installed above the ring, which scraped off the powder to a specific height such that powder of a consistent shape and height lay on top of the ring. Compressed air was then flowed over the ring at the ‘extraction site’, aerosolizing the powder and sucking it into the flow rig as the ring continuously rotated. By controlling the rotation speed of the ring and the height of the powder on the dosing ring, the aerosol mass flow rate was controlled by the operator. A 5 bar head pressure was developed, we started aerosolizing the powder once the differential pressure and flow rate stabilised through the clean filter.

TiO${}_{2}$ nanopowder was loaded into the reservoir of the dust generator. TiO${}_{2}$ was selected because it has similar mobility diameter, size distribution and packing density to soot; however, it is more attenuating to X-rays due to its higher atomic mass, hence making it easier to image with X-rays [24,25,26]. The particle mobility diameter distribution of the TiO${}_{2}$ had a peak at 25 nm and a tail of agglomerates in the order of 100 nm. This is comparable to engine out soot which in all but the most extreme operating conditions will have a peak diameter size distribution <100 nm and a tail of agglomerates in the order of 100 nm [26].

### 2.3. Sample Preparation

Samples were cut from an uncoated asymmetric SiC Diesel Particulate Filter (DPF) monolith. The samples were 3 cm long in the flow direction and contained a 3×3 cross section of channels as shown in Figure 3. The eight outer channels were blocked at the top of the sample using a quick set epoxy resin, shown in gold in Figure 3. The bottom of the central channel was blocked in the same manner. The sample was then glued into place in the sample holder using an epoxy resin. Using a 3×3 channel array means that the flow through the inlet channel walls to the outlet channels may be more realistic than studying through-wall-flow normal to a single wall only. Furthermore, having the channels surrounding the central channel means that the central channel is less likely to be damaged during cutting and handling, which could damage the pore structure being studied.

### 2.4. Data Processing

Projections were reconstructed using the filtered back projection algorithm into a volume using the Savu platform [27]. A distortion correction algorithm was applied to unwarp the images. The in-situ run lasted 20 min. The first scan was acquired at 2 min, after which a new scan was started every minute. Each volume in the sequence is then cropped to a 1410×1410×2110 voxel volume and centred on the centre of the central channel. The 2× objective lens used had an effective pixel size of 1.6 μm/pixel at the camera, thus this pixel volume represents a real volume of 2256×2256×3376μm. The volumes were then stacked as a time sequence. In the case of the in-situ run presented here, the data have the overall dimensions of 18×2256×2256×3376 (t, x, y, z) in 16 bit greyscale.

The volumes were then segmented. The volumes were converted to 8 bit and smoothed using a non-local means method. We created a mask of the solid filter in the first volume in the time series using a global threshold (pixels with greyscale values greater than 120 were included in the mask). The mask was then eroded and subsequently dilated. This removed noise and filled holes in the solid filter segment as desired. This mask was then registered onto the subsequent volumes in the times series. We then segmented the resulting masked volumes into a deposit segment and a background segment for each volume in the time series. In order to segment the deposit from the background, a manual threshold was applied globally across the entire time series (pixels with greyscale values greater than 112 were included in the deposit segment). This resulted in binary volumes of the deposit segment. Noise was removed by smoothing and then re-segmenting the deposit segment consistently across the entire data series. The filter and TiO${}_{2}$ deposit volumes were then combined for a fully labelled volume with filter, TiO${}_{2}$ deposit, and background segments, see Figure 4.

Finally a sensitivity analysis was performed for both thresholds to estimate uncertainty. Repeated segmentation shows that we can repeatedly select a threshold accurate to one grey level in an 8 bit image. A change in one above and below both chosen thresholds results in a volume being selected/deselected. For example, for a 600×600×184 voxel wall volume at minute 5 of the in-situ run ±0.75% of the volume is selected/deselected as a result of changing the thresholds by 1 either side of their chosen values.

## 3. Results

### 3.1. Visualising Filtration in a Wall Flow Filter

Figure 4 shows segments of the TiO${}_{2}$ deposit phase from the filter and void phase as TiO${}_{2}$ builds-up during the filtration. Figure 5 presents 3D volumes which visualise the build up of TiO${}_{2}$ deposits within the porous structure of the filter wall. The volumes encompass the inlet surface (top) of the filter wall to the outlet surface (bottom). Inspection of the segmented and greyscale images shows that TiO${}_{2}$ deposits are accurately segmented both in deep bed pores and on the surface of the channel wall, see Figure 4.

The technique’s spatial and temporal resolution make it possible to observe pore scale phenomena in the micro-CT images. Figure 4 shows the onset of a pore filling event. The pore indicated with a red circle is filled with TiO${}_{2}$ deposit during the filtration. The green arrow indicates a pore throat at the channel surface becoming choked by the deposit cake. The transition from deep bed filtration to cake filtration is resolved. Deep bed filtration occurs in the first 10 min of filtration when TiO${}_{2}$ deposits appear inside the filter wall pores, as shown by the red and green segments in Figure 5. TiO${}_{2}$ nanoparticles deposit in the deep bed pores because they are much smaller than the pores (10–30 μm pore diameter). Hence, when the filter is clean, deposits are filtered by inertial deposition, diffusion and direct interception rather than by sieving [2]. During the first 10 min caking behaviour is also observed in some areas of the channel surface, most noticeably in the inlet channel corners, as shown in Figure 4. After 10 min of loading, the TiO${}_{2}$ deposit in a caking layer only. The caking layer prevents deposit from reaching the deep bed by efficiently filtering particles. Therefore deposits only form on the top surface, shown by the blue segment in Figure 5d, and by the thick white TiO${}_{2}$ cake layer in Figure 4.

### 3.2. Quantifying Filtration in a Wall Flow Filter

A Representative Elementary Volume (REV) is a volume that is statistically representative of a bulk material property and has become an important characterization criteria for porous media [28,29,30,31]. We have analysed volumes that are representative of the average porosity in filter wall cross sections. During a filtration, porosity is determined not only by the porosity of the clean filter but also by the volume of deposit within each sub-volume. Pixel counting is used to calculate the porosity of sub-volumes of different sizes. Each volume contains the entire wall thickness of 295 μm (184 pixels), a height of 960 μm (600 pixels), but varying length. Figure 6a shows a REV cross sections in red. This cross sectional area is constant throughout all the volumes as we only change the length dimension between different volumes. We choose this set of dimensions because we know that porosity will change as a function of depth into the wall thickness due to deposition. By including the entire wall thickness in the subvolumes, we do not consider the porosity heterogeneity with depth. Hence, we can determine the length of filter wall required for the sample to be representative of the averaged porosity in a filter wall.

Figure 6 compares the porosity in volumes of different lengths and at different times during the filtration in order to determine REV size. The loading times tested span the runtime of the filtration at: t = 3 min, t = 11 min and t = 20 min. Five sub-volume sizes are compared during the REV analysis: ‘V600’ of size 960×960×295μm (600×600×184 voxels), ‘V400’ of size 640×960×295μm (400×600×184 voxels), ‘V200’ of size 320×960×295μm (200×600×184 voxels), ‘V050’ of size 80×960×295μm (50×600×184 voxels) and ‘V001’ of size 1.6×960×295μm (1×600×184 voxels). The porosity of 6 subvolumes samples was calculated for each subvolume size. The mean and variance is plotted in Figure 6 and recalculated for the three time points from the same 6 sub-volumes. Figure 6 shows that the ‘V400’ and ‘V600’ volumes are representative of average filter wall porosity as the mean filter wall porosity value plateaus across the time series. For further analysis we chose the ‘V600’ volume as it is still practical to compute and has lower standard deviation from the mean (1SD < 0.01).

We calculated porosity in a representative volume for 18 time points in the series (see Figure 7) in order to quantify the change in average deep bed porosity due to TiO${}_{2}$ deposition in a statistically representative fashion. The porosity value changes from an initial clean porosity of 0.60 to a final porosity of 0.56. We also use the V600 representative volume to calculate deep bed filtration rate. Filtration rate can be expressed as the rate of change with time of specific deposit volume;
$$N=\frac{\delta \sigma }{\delta t}=\frac{\left(V_{2}-V_{1}\right)\cdot {\left(REV\right)}^{-1}}{t_{2}-t_{1}}$$
where *N* is the filtration rate, σ is specific deposit volume, *t* is filtration time and $V_{1,2}$ is the volume of deposit in a representative volume (REV) at $t_{1,2}$. The peak filtration rate in the V600 volume occurs between 5–6 min, filtration rate then decreases until 10 min and then tends to values ∼0 after 10 min. This is indicative of the three filtration regimes described by Choi and Lee: deep bed filtration, transition filtration and cake filtration [13]. The peak filtration rate of 4 μm${}^{3}$·mm${}^{-3}$·min${}^{-1}$ occurs between 5–6 min. Showing the deep bed filtration rate is greatest at the beginning of the filtration (called the deep bed regime). Next the deep bed filtration rate decreases due to mixed behaviour during the transition regime, i.e., some cake filtration occurs lowering the deep bed filtration rate. Finally, the deep bed filtration rate tends to zero during the cake filtration regime. Cake filtration implies that after 10 min the deposit cake acts as a highly efficient filter which prevents any significant deep bed deposition from occurring inside the filter wall. This is plotted for different filter depths in Figure 8 (where filtration rate is $\frac{\delta y}{\delta z}$).

The V600 volume was also used to quantify penetration depth into the deep bed pores of the filter. In order to achieve this the 960×960×295μm (600×600×184 voxels) volume was divided into 18 sections, each with a shape of 960×960×16μm (600×600×10 voxels) width-wise, i.e., each sub-volume within the REV has a width of 16 μm (10 pixels) in the through wall flow direction (the final downstream subvolume has a width of 14 pixels). We calculate the volume percent of TiO${}_{2}$ in each of these sub-volumes by a pixel count of the TiO${}_{2}$ segment in each sub-volume and dividing by the total sub-volume. TiO${}_{2}$ deposit concentrations at different penetration depths are calculated for sub-volumes at different filtration times, as shown in Figure 8. In Figure 8, a depth of zero corresponds to the inlet surface; thus, we only consider the build-up of TiO${}_{2}$ deposits in the filter wall pores (i.e., deep bed), without considering the filter cake. We show that the concentration of TiO${}_{2}$ deposits decays with distance from the inlet channel surface. This decaying profile with depth is consistent throughout the filtration.

### 3.3. On-Line Differential Pressure Measurement

The pressure loss across the sample filter is plotted below in Figure 7. After 5 min, the pressure loss through the filter increases sharply, correlating with the start of deep bed pore filling events resolved in the images. Thus, in Figure 7, the representative volume porosity value drops just as the pressure loss spikes, showing good correlation between the pressure reading and image-based porosity plot. However, after 6 min the curve plateaus and then unexpectedly drops after 10 min. The pressure loss was caused by the HEPA filter loading with TiO${}_{2}$ that bypassed the filter sample and flowed between the sample holder walls and sample. The bypass increased backpressure in the flow rig and caused this artefact in the pressure data (The HEPA filter was installed at the back of the flow rig to prevent contamination of the experimental area).

## 4. Conclusions

We have demonstrated time resolved pore scale 4D microscopy of aerosol filtration for the first time. The technique allows for pore-scale phenomena occurring within an operational wall flow filter to be directly observed as they occur in a filter volume. We have demonstrated sufficient resolution to identify the locations of pore filling, pore blocking and filter surface caking events, alongside correlated differential pressure measurements across the filter. The TiO${}_{2}$ deposit was imaged with sufficient contrast compared to the surrounding SiC filter to allow accurate segmentation for image analysis. This revealed dynamic changes in porosity, filtration rate and deposition depth in statistically relevant subvolumes in the deep bed of an aerosol filter for the first time. Furthermore, we were able to observe the filter move from the deep bed filtration regime, through to the transition and cake filtration regime, in good agreement with literature.

This methodology provides opportunities for new insights into the mechanisms by which filter efficiency and pressure loss can be affected in service. These in-turn determine the concentration of PM output from the exhaust and the fuel efficiency of the vehicle power unit. Furthermore, it will allow filtration simulations to be validated against experimental 4D data of the pore space which has not previously been possible. We have demonstrated this method on a relatively simple single phase wall flow filter with an idealized particulate matter. Nonetheless, this method could be directly applied to catalyst coated automotive wall flow filters in order to dynamically image the interface between the soot like deposits and the catalyst layer. Furthermore, we believe this method could be adapted to have broader application in aerosol filter characterization. Aerosol filters are applied broadly across many different areas from heating, ventilation and air conditioning (HVAC) applications, to industrial air intake units, clean rooms and PPE (such as respirators that protect against bio-aerosols).

The principal experimental considerations for imaging aerosol filtration by this method are: (1) the degree of similarity between the model powder and the real world particulate matter, (2) the ability of the dust generator to aerosolize the powder, (3) the transport properties of the airborne particulate though the connecting tubing, (4) the degree of contrast available using the X-ray spectrum able to penetrate the sample holder, (5) the ability to generate a realistic volumetric flow rate through the filter and (6) the spatial and temporal resolution of the imaging system. Efforts to extend these capabilities are ongoing.

## Figures and Tables

**Figure 1 materials-13-05676-f001:**
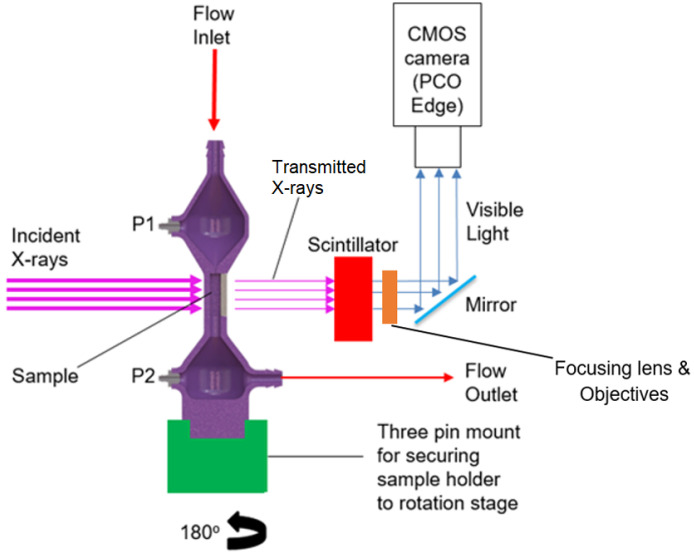
Imaging set-up for fast synchrotron imaging at Diamond Light Source beamline I13-2. Aerosol is flowed continuously through the 3D printed sample holder (purple) from top to bottom whilst it is rotated through 180${}^{\circ }$ for tomography.

**Figure 2 materials-13-05676-f002:**
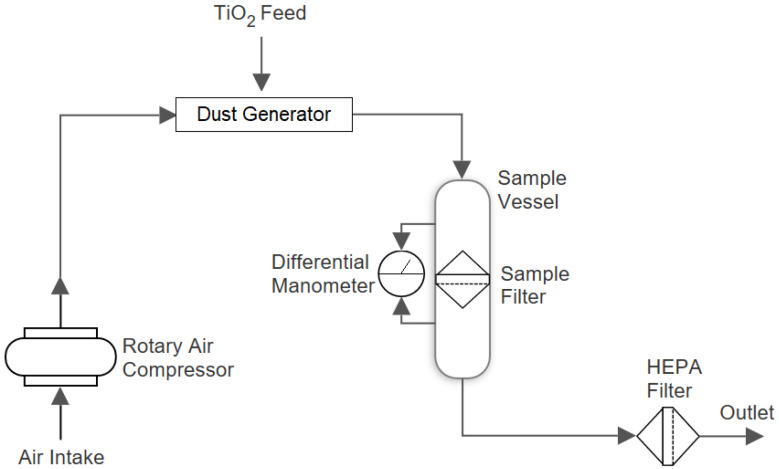
Diagram showing the components used in the flow rig set-up in I13-2: the air compressor generates the head pressure used to flow the aerosol through the rig. TiO${}_{2}$ is aerosolized in the dust generator by flowing the compressed air over a dosing ring that holds the nanopowder. A differential digital manometer is used to measure the pressure loss across the filter as the aerosol flows through the sample holder. A High Efficiency Particle Air (HEPA) filter is used as an exhaust filter to prevent contamination of the hutch environment.

**Figure 3 materials-13-05676-f003:**
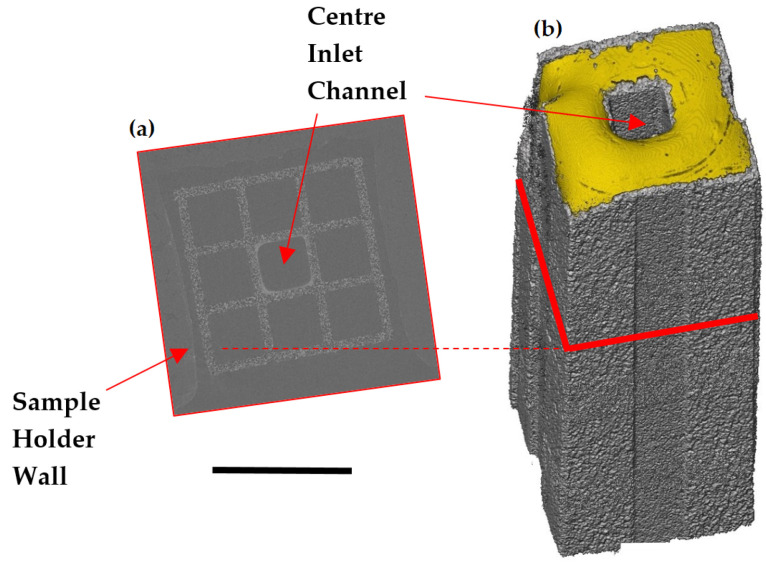
Micro-CT slice (**a**) and a micro-CT volume (**b**) showing a typical wall flow filter sample. Samples cut from the monoliths were 2–3 cm long, with 3×3 channels, as shown in the slice image. Aerosol entered the filter through the centre channel, indicated in the figure, and was prevented from entering the exterior channels which were plugged with epoxy resin, shown here in gold. The centre channel is blocked in the same way at the downstream end. Scale bar = 4 mm.

**Figure 4 materials-13-05676-f004:**
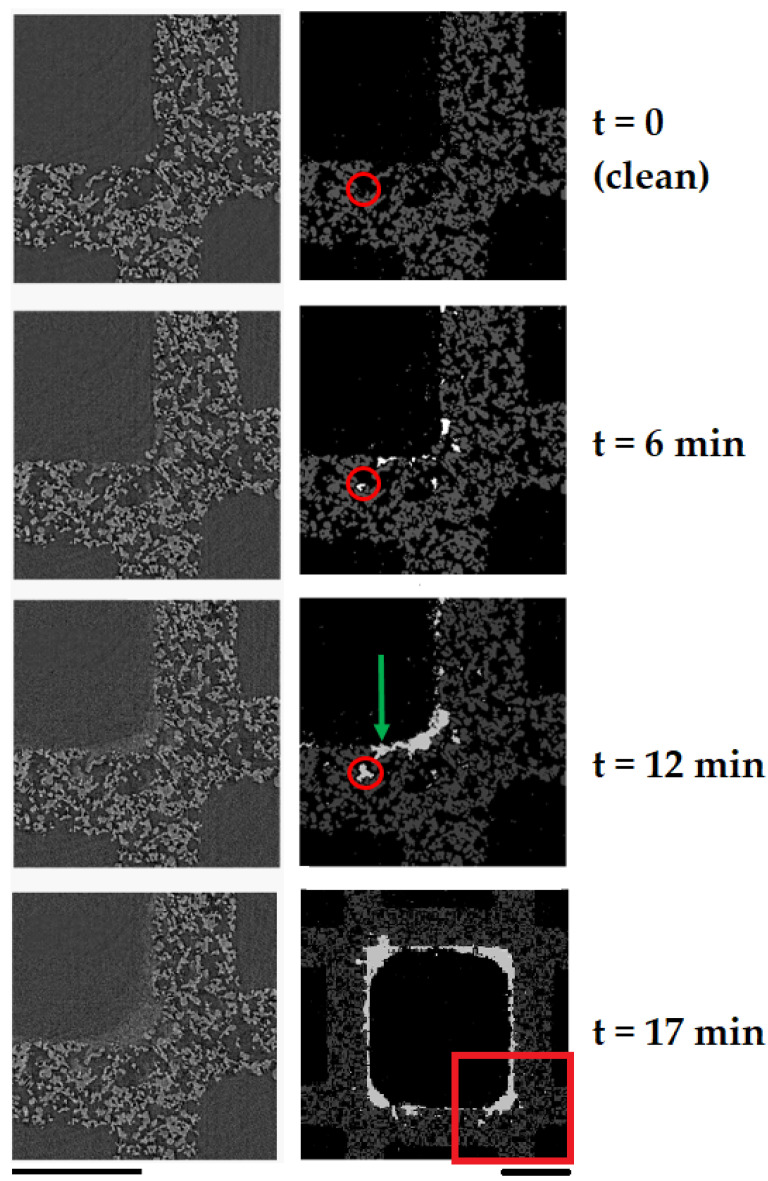
Comparison of greyscale micro-CT virtual slices (left column) and corresponding segmented slices (right column). Slices are 8 mm from the top (upstream) inlet of the sample. TiO${}_{2}$ aerosol deposits are white, the filter segment is grey and the pore and background segments are black. The red circle tracks a pore filling event that occurred during the time series. The green arrow indicates a pore blocking event caused by the growing cake layer. The final image indicates the position of the previous slices within the channel. Scale bars = 500 μm.

**Figure 5 materials-13-05676-f005:**
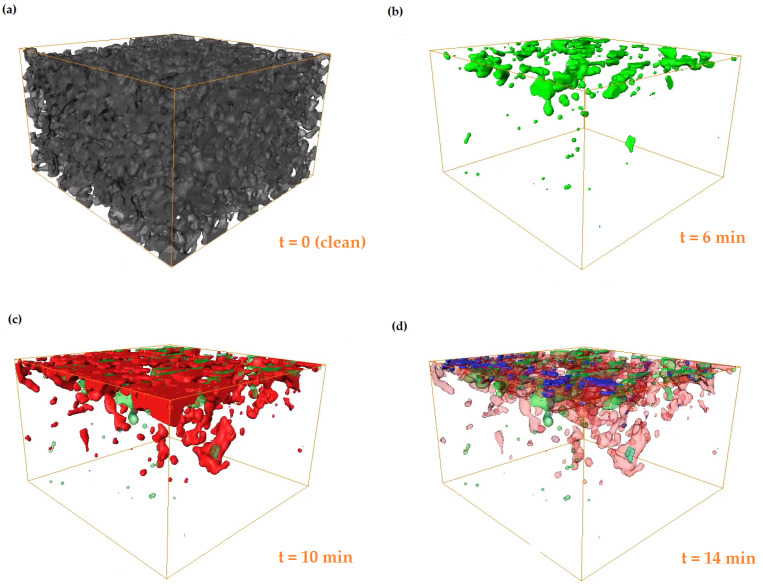
480×480×295μm (300×300×184 pixels) micro-CT volume showing TiO${}_{2}$ deposits in the pore space at different times during the filtration, the aerosol flow is from top to bottom. The volume is a quadrant from the REV indicated in Figure 6a. (**a**) t = 0 (showing the rendered filter phase only, shaded in grey), (**b**) 6 min (green), (**c**) 10 min (red) and (**d**) 14 min (blue).

**Figure 6 materials-13-05676-f006:**
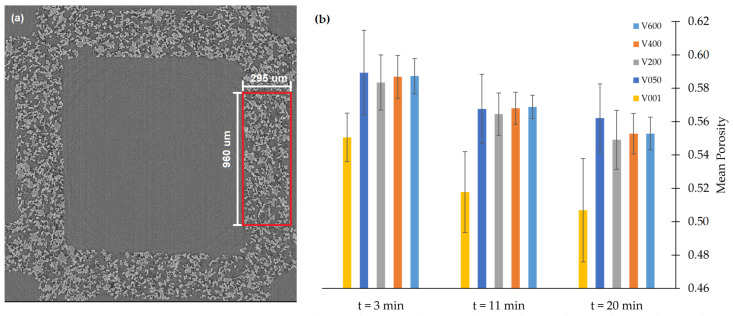
(**a**) A virtual micro-CT slice showing a cross section of a volume used for REV determination. (**b**) Plot of the mean porosity and error (±1SD) for five different sub-volume lengths (V600, V400, V200, V050 and V001) at three different loadings (t = 3 min, t = 11 min, t = 20 min).

**Figure 7 materials-13-05676-f007:**
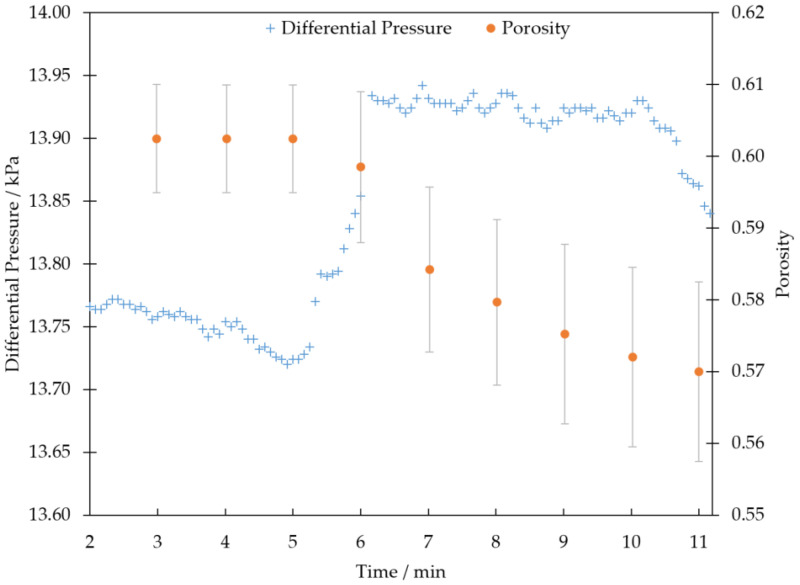
Plot of porosity and the pressure difference between the filter sample inlet and outlet against time during an in-situ filtration. Each porosity data plot represents the point in the time series where a tomography scan was finished (∼1 per minute), each porosity value is calculated from a representative volume as discussed above. Porosity error bars are calculated as described in Section 2.4 and the differential manometer has a resolution of 0.01 kPa.

**Figure 8 materials-13-05676-f008:**
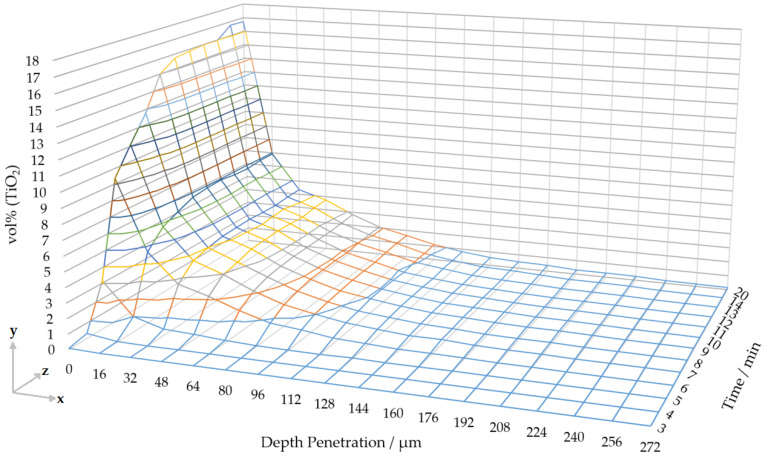
Concentration of TiO${}_{2}$ deposits (*y*-axis) against penetration depth (*x*-axis) and time (*z*-axis) for porosity in a representative volume during continuous filtration. The volume was divided into 16 μm wide sub-volumes along the width of the filter wall and TiO${}_{2}$ volumes calculated from within each of these sub-volumes in the time series, these are expressed as volume percentages in the figure.

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
