# Peer review of "4D In-Situ Microscopy of Aerosol Filtration in a Wall Flow Filter"

_materials, 2020, doi:10.3390/ma13245676_

Round 1

Reviewer 1 Report

This is a referee report for the manuscript “4D in-situ microscopy of aerosol filtration in a wall flow filter” submitted to MDPI Materials.

The paper reads well, and surely is of interest for the X-ray micro-tomography community.

I have only minor remarks:

Figure 5: The numbers giving the time in the images is poorly readably, and (c) states 600 s in the image and 540 s in the caption. Please improve and correct.

Figures 6 and 8: Please increase the font size of the axis labelling for better readability, and improve the resolution of the blue curve in Fig.8.

Author Response

The paper reads well, and surely is of interest for the X-ray micro-tomography community. I have only minor remarks:

  1. Figure 5: The numbers giving the time in the images is poorly readably, and (c) states 600 s in the image and 540 s in the caption. Please improve and correct. Response: Edited correction into manuscript. Caption -> “10 minutes” (600 s); Figure 5 -> Changed font and increased font size to be more readable.
  2. Figures 6 and 8: Please increase the font size of the axis labelling for better readability, and improve the resolution of the blue curve in Fig.8. Response: Edited correction into Manuscript. Changed the style of the blue line plot so that experimental data points are visible. Increased font size of labels to be more readable.

Reviewer 2 Report

This paper presents the 4D (time-resolved) tomography study of aerosol filtration in a wall flow filter. It considers dynamics of aerosol depositions in such filter and its porosity.

According to the text of the manuscript the following remarks can be done.

  1. Line 76. Please, add gaps between number and units. Also, please, check this throughout the manuscript, as far as these misprints occurs several times (lines 106-109, 152, etc.)
  2. Line 80. Please provide reference and information about exactly model of CMOS camera you have used.
  3. Lines 103-106. “TiO2 was selected because it has similar mobility diameter, size distribution and packing density to soot, however it is more attenuating to x-rays due to its higher atomic mass, hence making it easier to image with x-rays”. Please, prove this statement with reference. Is it possible to estimate how much time one needs to make the same quality images for natural soot. Is contrast resolution of proposed method enough for that?
  4. Lines 107-109. “This is comparable to engine out soot which in all but the most extreme operating conditions will have a peak diameter size distribution <100nm and a tail of agglomerates in the order of 100nm”. Please, prove this statement with reference.
  5. Line 113 and Figure 3 caption. Please, use the same color identification, either gold or yellow.
  6. Lines 124-129. Authors say that one in-situ run lasted approximately 20 minutes, and tomographic scan takes one minute, so about 20 volumes is obtained for each run. However, time dimension of experimental data determined as 18. What is the reason for this difference? Does it result of uncertainty for the each scan measurement time or delays between scans? Please, clarify this point.
  7. Lines 130-131. Please, clarify how you smooth converted volume and create filtration mask. What was global threshold that you have used?
  8. Figure 4. What is the depth (distance along the channel from the top) at which this slices is obtained? Does it have sense to add in figure another set of segmented slices for another depth? Please, provide the figure with better time captions.
  9. Figure 5. In caption figure 5.c identified as TiO2 deposition in the pore space after 540 seconds, while in figure and text authors speak about 600 seconds. Please, check this difference.
  10. Figure 5. Please, identify in the text what exactly volume you consider in here. Is it volume allocated in the green arrow direction with specified dimensions?
  11. Page 8. In description of Figure 6, authors use for spatial sizes both μm (lines 168-169, Figure 6.a) and pixel (lines 179-180, Figure 6.b) scales that is a little confusing. Please, consider the possibility to use one scale. For example, provide μm values with corresponding pixel values.
  12. Figure 6.b. As it follows from experimental description, authors obtained a set of images (one CT scan) each 60 second. It means that considered time coordinates should be multiples of 60 seconds. However, first point considered by authors is 170 second. Is that a misprint? Considering this remark, I would recommend to use time scale in minutes, as far as time resolution in study is 1 minute. Using of seconds as units for this scale is a little bit confusing and I do not see any necessity to use it. For example, considering filtrations regime, one cannot really say, that “Deep bed filtration occurs in the first 600s”, and not in 590 s. But it is possible to conclude, that this filtration occurs in the first 10 minutes, and not in 9. Please, consider the possibility to use minute scale.
  13. Figure 8. Does not it better to use points for pressure plot rather than continuous line? Using of splines between experimental points is not entirely correct. Besides, it will be nice to provide experimental points with measurement errors.
  14. Lines 176-184. “Figure 3 shows that we have achieved a REV for the ’V600’ (600×600×184 pixels) volume as the mean porosity value has plateaued and the variance has reduced to acceptable values (> 0.02) across the time series”. This figure clearly shows, that mean porosity for all sub-volume lengths is statistically the same, excepted V001. So it is not clear, why authors consider ’V600’ as a REV rather than ’V400’ or ’V200’. The only reason mentioned in text is that “the variance has reduced to acceptable values (> 0.02)”. But why authors consider 0.02 as an “acceptable value”? For example, variance for’V400’ is almost the same. Please, clarify this. Besides, probably authors should use “Figure 6” instead of “Figure 3” (line 182) and “< 0.02” instead of “> 0.02” (line 184).

In conclusion, I would like to note that the text of the manuscript contains several misprint and would recommend to the authors carefully check the text. Provided that the above suggestions are taken into account, the paper can be recommended for publication.

Author Response

According to the text of the manuscript the following remarks can be done.

  1. Line 76. Please, add gaps between number and units. Also, please, check this throughout the manuscript, as far as these misprints occurs several times (lines 106-109, 152, etc.)

Response: Edited corrections into manuscript.

  1. Line 80. Please provide reference and information about exactly model of CMOS camera you have used.

Response: Edited manuscript to include exact model of camera. Edited Manuscript Line 80 -> “to project visible light through the objective to a CMOS camera (pco.edge 5.5 ) to capture radiographic projections”

  1. Lines 103-106. “TiO2 was selected because it has similar mobility diameter, size distribution and packing density to soot, however it is more attenuating to x-rays due to its higher atomic mass, hence making it easier to image with x-rays”. Please, prove this statement with reference. Is it possible to estimate how much time one needs to make the same quality images for natural soot. Is contrast resolution of proposed method enough for that?

Response: Edited Manuscript -> See reference 24-26 (SMPS of TiO2 and Soot)

We do not believe enough contrast can be achieved from soot in this in-situ rig.

  1. Lines 107-109. “This is comparable to engine out soot which in all but the most extreme operating conditions will have a peak diameter size distribution <100nm and a tail of agglomerates in the order of 100nm”. Please, prove this statement with reference.

Response: Edited Manuscript -> See Reference 26 (SMPS/ELPI study of soot)

  1. Line 113 and Figure 3 caption. Please, use the same color identification, either gold or yellow.

Response: Edited Manuscript. Line 113 -> “a quick set epoxy resin, shown in gold in Figure 3”

  1. Lines 124-129. Authors say that one in-situ run lasted approximately 20 minutes, and tomographic scan takes one minute, so about 20 volumes is obtained for each run. However, time dimension of experimental data determined as 18. What is the reason for this difference? Does it result of uncertainty for the each scan measurement time or delays between scans? Please, clarify this point.

Response: The first tomography scan finishes at 2 minutes (120 s). Hence 18 tomograms in the 20 minute in-situ run reported here. Many samples (that are not reported here) were scanned, the average number of tomograms acquired was 20, hence the erroneous inclusion of this 20 volumes value in the original manuscript. We will remove this reference to data not reported in this article and edit to clarify.

Edited Manuscript Line 122 – 129 -> “A distortion correction algorithm was applied to unwarp the images. The in-situ run lasted 20 minutes. The first scan was acquired at 2 minutes (120 s), after which a new scan was started every minute. Each volume in the sequence is then cropped to a 1410 x 1410 x 2110 pixels volume and centered on the center of the central channel. The 2x objective lens used had an effective pixel size of 1.6 um/pixel at the camera, thus this pixel volume represents a real volume of 2256 x 2256 x 3376 um. The volumes were then stacked as a time sequence. In the case of the in-situ run presented here the data has the overall dimensions of 18 x 2256 x 2256 x 3376 (t, x, y, z) in 16 bit grayscale.”

  1. Lines 130-131. Please, clarify how you smooth converted volume and create filtration mask. What was global threshold that you have used?

Response: Edited Manuscript. Line 130 – 132 -> “The volumes were then segmented. The volumes were converted to 8 bit and smoothed using a non-local means method. We created a mask of the solid filter in the t = 0 volume using a global threshold (pixels with greyscale values greater than 120 were included in the mask).”

  1. Figure 4. What is the depth (distance along the channel from the top) at which this slices is obtained? Does it have sense to add in figure another set of segmented slices for another depth? Please, provide the figure with better time captions.

Response: Slices in Figure 4 are 8 mm from the top of the channel. We edited in this information to the Fig 4 caption. Fig 4 has been edited to have better quality labels. However, we believe adding another set of 2D slices does not add anything to the analysis as they are visually extremely similar and different depths of filter can be observed in the volumes in fig 5.

  1. Figure 5. In caption figure 5.c identified as TiO2 deposition in the pore space after 540 seconds, while in figure and text authors speak about 600 seconds. Please, check this difference.

Response: Edited Manuscript; “10 minutes”

  1. Figure 5. Please, identify in the text what exactly volume you consider in here. Is it volume allocated in the green arrow direction with specified dimensions?

Response: The volume is the lower upstream quadrant from the REV indicated in Figure 6.

Edited Manuscript: Figure 5. Caption -> “480x480x295 mm micro-CT volume showing TiO2 deposits in the pore space at different times during the filtration, the aerosol flow is from top to bottom. The volume is a quadrant from the REV indicated in Figure 6.”

  1. Page 8. In description of Figure 6, authors use for spatial sizes both μm (lines 168-169, Figure 6.a) and pixel (lines 179-180, Figure 6.b) scales that is a little confusing. Please, consider the possibility to use one scale. For example, provide μm values with corresponding pixel values.

Response: In the results section we have edited to the format -> “X µm (Z pixels)”

  1. Figure 6.b. As it follows from experimental description, authors obtained a set of images (one CT scan) each 60 second. It means that considered time coordinates should be multiples of 60 seconds. However, first point considered by authors is 170 second. Is that a misprint? Considering this remark, I would recommend to use time scale in minutes, as far as time resolution in study is 1 minute. Using of seconds as units for this scale is a little bit confusing and I do not see any necessity to use it. For example, considering filtrations regime, one cannot really say, that “Deep bed filtration occurs in the first 600s”, and not in 590 s. But it is possible to conclude, that this filtration occurs in the first 10 minutes, and not in 9. Please, consider the possibility to use minute scale.

Response: Edited manuscript so that time dimension is in minutes.

  1. Figure 8. Does not it better to use points for pressure plot rather than continuous line? Using of splines between experimental points is not entirely correct. Besides, it will be nice to provide experimental points with measurement errors.

Response: Edited manuscript to include experimental points. Error bars clutter the graph too much to be included on the pressure reading. We include the resolution of the manometer in the caption and added error bars to the porosity plot based on a sensitivity analysis described in section 2.4.

14. Lines 176-184. “Figure 3 shows that we have achieved a REV for the ’V600’ (600×600×184 pixels) volume as the mean porosity value has plateaued and the variance has reduced to acceptable values (> 0.02) across the time series”. This figure clearly shows, that mean porosity for all sub-volume lengths is statistically the same, excepted V001. So it is not clear, why authors consider ’V600’ as a REV rather than ’V400’ or ’V200’. The only reason mentioned in text is that “the variance has reduced to acceptable values (> 0.02)”. But why authors consider 0.02 as an “acceptable value”? For example, variance for’V400’ is almost the same. Please, clarify this. Besides, probably authors should use “Figure 6” instead of “Figure 3” (line 182) and “< 0.02” instead of “> 0.02” (line 184).

Edited manuscript: Lines 184 -187 -> Figure 6 shows that the 'V400' and 'V600' volumes are representative of average filter wall porosity as the mean filter wall porosity value plateaus across the time series. For further analysis we chose the 'V600' volume as it is still practical to compute and has lower standard deviation from the mean (1SD < 0.01).

In conclusion, I would like to note that the text of the manuscript contains several misprint and would recommend to the authors carefully check the text. Provided that the above suggestions are taken into account, the paper can be recommended for publication.

Round 2

Reviewer 2 Report

This study is a worth publication and can be published in the journal Materials.

Author Response

Thanks

This manuscript is a resubmission of an earlier submission. The following is a list of the peer review reports and author responses from that submission.